# Revealing the History and Mystery of RNA-Seq

Aishwarya Gondane * and Harri M. Itkonen *

Department of Biochemistry and Developmental Biology, Faculty of Medicine, University of Helsinki, 00014 Helsinki, Finland
* Correspondence: aishwarya.gondane@helsinki.fi (A.G.); h.m.itkonen@gmail.com (H.M.I.)

**Abstract:** Advances in RNA-sequencing technologies have led to the development of intriguing experimental setups, a massive accumulation of data, and high demand for tools to analyze it. To answer this demand, computational scientists have developed a myriad of data analysis pipelines, but it is less often considered what the most appropriate one is. The RNA-sequencing data analysis pipeline can be divided into three major parts: data pre-processing, followed by the main and downstream analyses. Here, we present an overview of the tools used in both the bulk RNA-seq and at the single-cell level, with a particular focus on alternative splicing and active RNA synthesis analysis. A crucial part of data pre-processing is quality control, which defines the necessity of the next steps; adapter removal, trimming, and filtering. After pre-processing, the data are finally analyzed using a variety of tools: differential gene expression, alternative splicing, and assessment of active synthesis, the latter requiring dedicated sample preparation. In brief, we describe the commonly used tools in the sample preparation and analysis of RNA-seq data.

**Keywords:** bioinformatics; transcriptomic data analysis; RNA-seq; alternative splicing; nascent mRNA analysis; scRNA-seq

## 1. Introduction: Evolution of Sequencing Technologies

The discovery of the double helical structure of DNA by Watson and Crick formed the basis of a new field of science focusing on the molecular biology of the cell at the ultimate backbone of life [1]. At the molecular level, the transfer of information from DNA to RNA to protein governs all the processes in the cell [2]. By measuring the mRNA levels, we can evaluate how cells remodel their transcriptome to adapt to the existing environment (for example health and disease).

The discovery of the first sequencing technique in 1975, Sanger sequencing, opened the door to understanding the dynamics of the genetic information [3]. In 1977, Maxim and Gilbert reported a novel technique of sequencing DNA by chemical degradation [4]. In the early days of sequencing, the experimental part took a long time, and the overall sequencing length was modest (some 100 base pairs). In 1988, automation of the Sanger sequencing offered a solution to both of these limitations and allowed sequencing lengths up to 500 base pairs [5]. Development of Sanger-sequencing eventually resulted in the first ever complete human genome sequence, which was accomplished through a collaborative initiative of 20 groups from around the globe in 2001 [6]. Moving closer to the modern day, automation of most of the steps enables the processing of multiple samples in parallel, decreasing the need for human intervention and the likelihood of mistakes. These advancements in automated sequencing techniques make it possible to quantitate gene expression levels across diverse samples.

In 1995, Schena et al. reported a method to quantitate gene expression levels using a small chip: the microarray [7]. For the first time, it was possible to analyze gene expression genome-wide. However, this technique could be used only for already known target genes. Another limitation of the microarray is the lack of exon-level information. This led

to the development of exon microarrays [8], thereby providing deeper insights into the underlying biology.

Finally, the discovery of the massively parallel sequencing techniques has significantly reduced the cost and time required to generate gene expression data across the entire transcriptome of a species. Datasets generated by massively parallel sequencing are large and require high-power computational resources; these needs gave rise to a new field of science: bioinformatics.

In the past 10 years, the development of sequencing technologies has led to an exponential increase in both the number and size of the datasets generated. At the same time, massive number of the new techniques require development of the computational tools to decode the biological significance. Here, we provide an overview on the history and mystery of RNA-seq (Figure 1).

### Overview of RNA-seq data analysis

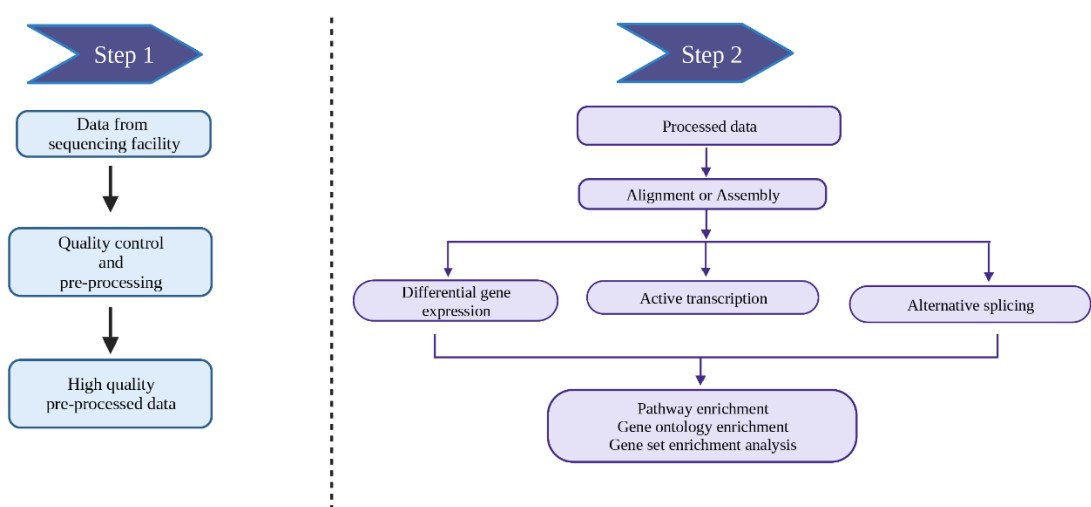

**Figure 1.** Overview of RNA-seq data analysis. In this review, we also describe the analysis of nascent RNA, alternative splicing, and scRNA-seq.

In the subsequent chapters, we first describe the basic tools to analyze RNA-seq data and consider the advantages and disadvantages of those (Table 1). Second, we move on to more complex RNA-seq analysis tools, including the assessment of active synthesis and alternative splicing. Third, we describe the tools to analyze single-cell RNA-seq data. When a new technique is introduced, we first briefly describe the wet lab experimental part and then move on to describe the computational tools. Finally, we propose where the field is heading in the future.

**Table 1.** Comparison of traditional RNA-seq, SLAM-seq, and scRNA-seq. The below sequencing technologies have advantages and disadvantages; selecting the right approach depends on the experimental setup.

| Technique | Description | Advantages | Limitations |
|---|---|---|---|
| Standard RNA-seq | Quantifies the levels of RNA from a biological sample at a given moment. | • Not limited to known genes [9]. <br> • Enables analysis of alternative splicing. | • Large amount of starting material needed (for typical library-preparation). <br> • Masks sample heterogeneity (for example in the case of biopsy [10]). |

**Table 1.** *Cont.*

| Technique | Description | Advantages | Limitations |
|---|---|---|---|
| Nascent RNA-seq | Nucleotide analogue based techniques used to assess RNA synthesis. The nascent RNAs incorporate the nucleotide analogue and are either enriched using affinity-based techniques or decoded computationally. | • Measures active transcription.<br>• Direct causality between regulator and target.<br>• Allows evaluation of mRNA half-lives and of mRNA degradation. | • Laborious compared to RNA-seq.<br>• Selecting appropriate labeling time is critical: if too short, fails to provide a snapshot of mRNA synthesis; if too long, nascent and total-RNA cannot be distinguished. |
| scRNA-seq | Measures the gene expression levels at single cell resolution. | • Allows assessment of the sample heterogeneity.<br>• Enables tracking (potential) cell-fate transitions/developmental steps. | • Laborious compared to RNA-seq.<br>• Getting the cells to a single cell solution is challenging and choosing the right technique to do this is crucial. |

## 2. Data Pre-Processing

The sequencing data are shared in the FastQ format by the sequencing facility. This format is a modified version of the standard fasta-format, and every read is described by four lines: the first line begins with "@" followed by the sequence identifier, the second line has the raw sequence, the third line is a "+", and the last line has the quality values corresponding to the raw sequence, the "phred" score. The pre-processing of the raw FastQ file consists of quality check, adapter removal, trimming, and filtering (Figure 2).

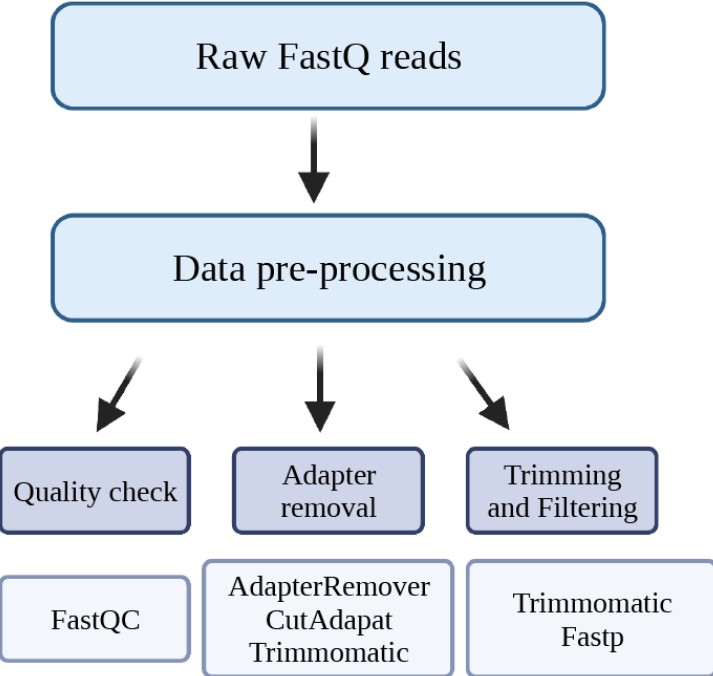

**Figure 2.** Data pre-processing steps. Checking the quality of the reads in the raw FastQ files is a crucial step in the sequencing data analysis pipeline. The quality of the reads is evaluated based on the phred score which is assigned to each nucleotide within the read. The higher the phred score is, the better the quality of the data. The results from the quality control dictate how much refinement of the data is required. In any case, adapters must be removed, and occasionally additional trimming is required.

### 2.1. Quality Check

Evaluating the quality of the FastQ files is crucial to correctly describe the biological significance of the results obtained. Low-quality reads in the FastQ files can arise from the adapter contamination on either side of the read or due to technical issues arising from the sequencer. Failure to filter the low-quality reads from the RNA-seq data can lead to severe issues including increased background and detection of false alternative splicing events. Essentially, the low-quality reads might not map to any region in the reference genome or might map to multiple regions, leading to low mapping quality. In the case of splicing analysis, if the length of the reads is different, some of the reads may incorrectly map to the intronic regions and thereby be called a splicing defect. FastQC, the most widely used Java-based software to evaluate the quality of FastQ files, is freely available from https://www.bioinformatics.babraham.ac.uk/projects/fastqc/ (accessed on: 23 February 2023). Analysis modules of the FastQC report can be used to confirm the quality of the data, or, alternatively, select the appropriate tools to trim the FastQ files prior to moving on to the actual data analysis. An important module in the FastQC report is the "per base sequence quality" (Figure 3A,B). The box and whisker plot in this module depicts the phred score (quality) of each nucleotide called at a given position. Phred score above 20 is acceptable. If the score is below 20, data have to be further pre-processed (see below). The analysis report is also used to evaluate potential adapter contamination and the read length distribution.

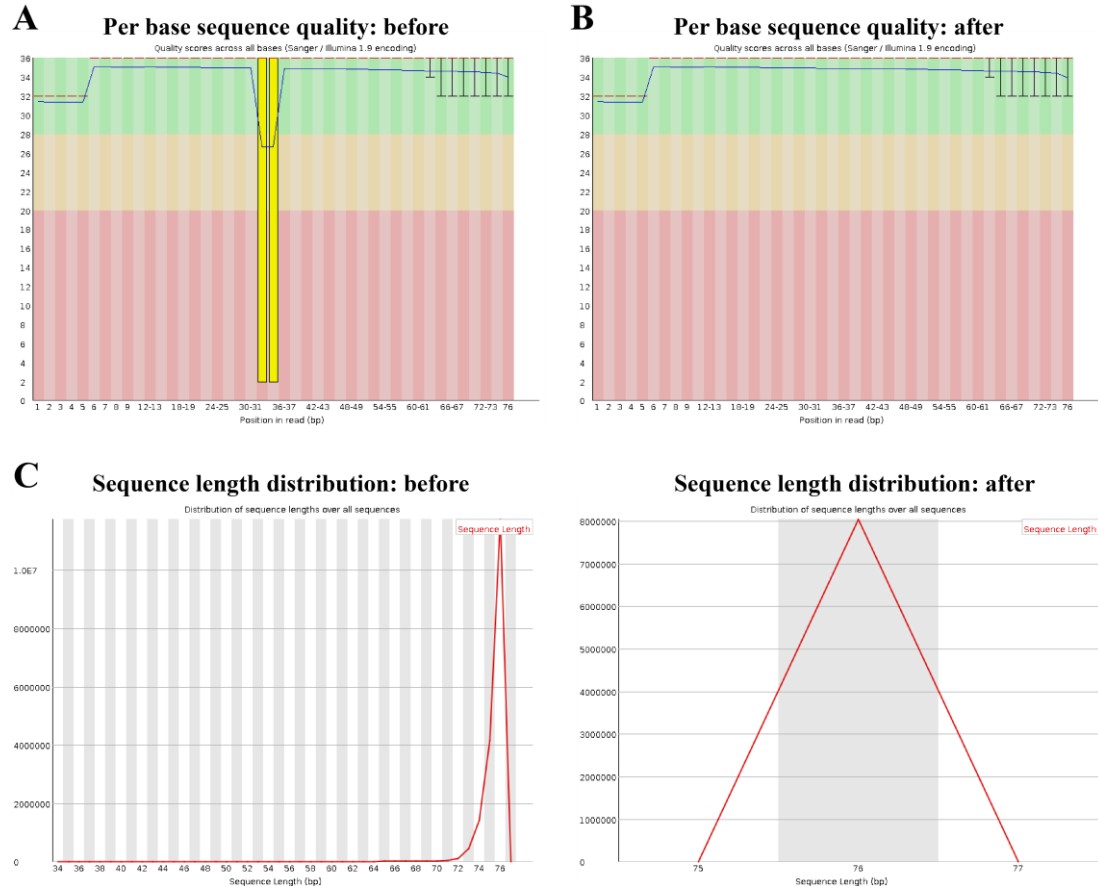

**Figure 3.** Evaluating and editing the FastQ files using trimmomatic. (**A**) The FastQC report shows that some bases across all of the reads are of low quality, i.e., the phred score is almost equal to 2. (**B**) Using trimmomatic, these low-quality bases can be removed from the FastQ file. (**C**) FastQC reports of the sample with varying read length before (**left**) and after the trimming (**right**). The line plot depicts that majority of the reads have read lengths ranging from 72 to 76, but shorter reads are also present. The short reads are completely removed after trimming (**right**).

## 2.2. Adapter Removal

Adapters are short oligonucleotides that must be removed before the analysis as they will interfere with the alignment to the reference genome. In the sequencing reaction, adapters allow the fragmented sample to bind to the lanes of the sequencer. The widely used tools for adapter trimming are: AdapterRemover https://adapterremoval.readthedocs.io/en/stable/ (accessed on: 23 February 2023) [11], CutAdapt https://cutadapt.readthedocs.io/en/stable/installation.html (accessed on: 23 February 2023) [12] and trimmomatic http://www.usadellab.org/cms/?page=trimmomatic (accessed on: 23 February 2023) [13]. AdapterRemover can remove adapters from 5′- and 3′-end, single- and paired-end data, but cannot be used to trim multiple adapters at once. On the other hand, CutAdapt can handle multiple adapters at the same time, but cannot be used for paired-end data. Trimmomatic is capable of trimming adapters from 3′-end of both single- and paired-end data, and handle multiple adapters, but fails to remove the adapters from the 5′-end. In brief, the choice of tool depends on the specific task requirements.

## 2.3. Trimming and Filtering

Trimming and filtering are used to trim the reads contributing to read length variation and to remove low-quality, uninterpretable reads. If there are only a few nucleotides failing the quality threshold, trimming using tools such as Trimmomatic [13] aids to maintain the sequencing depth whilst discarding the low-quality reads (compare Figure 3C left and right). However, when entire reads must be filtered out, software such as fastp is more suitable as it allows filtering the reads based on the quality and length threshold [14]. Fastp can be accessed freely from here: https://github.com/OpenGene/fastp (accessed on: 23 February 2023).

## 3. Data Analysis

Now that the data are of high quality, we can move on to the main analysis. In this section, we describe the tools to analyze differential gene expression, alternative splicing, nascent mRNA synthesis, and single-cell RNA-seq (scRNA-seq) data.

### 3.1. Alignment

The first step in RNA-seq data analysis is mapping the reads from the raw FastQ file and generating the putative transcriptome. When the reference genome of the organism-of-interest is available, reference-based mapping is used. If the reference genome is not known, de novo assembly is used. In this case, the short reads are merged to form the contig, "a hypothetical genome", to which the same reads are re-mapped. The alignment data are stored in the Sequence Alignment Map (SAM) file. This file has 11 mandatory columns and might contain several optional columns. To economize the storage space and accelerate the downstream processing, the alignment files are converted to their binary form (BAM; binary alignment map). Samtools is used to read and manipulate SAM and BAM files [15].

Based on the availability of the reference genome, the alignment algorithm is chosen. The reference-guided assembly can be completed using TopHat [16], STAR [17], or Bowtie [18], whereas Trinity [19] is a robust algorithm used for efficient reference-free mapping. Some of the aligners generate temporary files during the alignment, which increases the storage space requirement, for example, STAR generates large temp files, while Bowtie does not. Downstream analysis may require a specific alignment tool to be used, typically when more complex data analysis is of interest. Such examples include: alternative splicing analysis using rMATS [20] and metabolically labeled RNA-seq data analysis using grandR [21], which both rely on STAR.

### 3.2. Differential Gene Expression Analysis

One of the most prevailing applications of RNA-seq is to study the changes in gene expression levels between two or more conditions. The first step is to remove the non-

uniformities in the samples, thereby ensuring the high quality of the samples to be analyzed. These non-uniformities are caused by signal decay as the sequencing reaction proceeds toward the 3′ end. The signal decay leads to inconsistent read coverage across the read length and can be resolved in the data pre-processing steps (see Section 2.3 and Figure 3C). Typically, this will not be an issue for the samples sequenced at the same time on the same flow cell; the non-uniformities "within" the sample are masked because the same bias is observed for all the samples in the sequencing run. However, these uniformities would be an issue if the data generated in different flow cell runs is to be compared; in these cases, a high number of replicates is required.

The differential gene expression analysis is performed between samples, and therefore the non-uniformities across different samples require normalization. A normalized expression unit is used to remove the technical non-uniformities from the sequencing data. Reads per kilobase per million reads mapped (RPKM) is the simplest normalization method. RPKM can be used for both single- and paired-end sequencing, and it corrects for the differences in both the library sizes and the gene length [22]. FPKM (fragments per kilobase of transcript per million mapped reads) is analogous to RPKM but it is used for paired-end sequencing data [23]. Normalization is particularly important when the samples vary in sequencing depth. This variation can be visualized by clustering the samples prior to normalization (principal component analysis), and by looking at the number of reads.

After normalization is complete, differential gene expression analysis can be performed. The most frequently used methods for calling the differentially expressed genes (DEGs) are DESeq2 [24] and edgeR [25]. DEG calling algorithms depend on the count of reads mapped to a genomic location. The count of mapped reads is presented in a form of a matrix, where each row represents genes from the reference genome and the columns are the reads mapped to that gene. These count matrices can be calculated using functions such as "summarize overlaps" in R or by using the featureCounts tool. FeatureCounts is a widely used method for computing the number of reads as it is accurate, fast, and easy to use [26]. DESeq2 uses the median of ratios to normalize read counts to account for sequencing depth and nucleotide composition, while edgeR uses "trimmed mean of mapped values". Both DESeq2 and edgeR normalize the samples to account for size differences and variance in the gene length. This normalization is part of the semi-automated DESeq2 pipeline, whilst, in edgeR, the user must perform the normalization. Additional tools for DEG calling include NBPSeq [27], which is based on negative binomial distribution; and the two-stage Poisson model (TSPM) [28], which can be used for analyzing RNA-seq data with small sample sizes. BaySeq utilizes empirical Bayesian analysis to identify the differentially expressed genes, and, like edgeR, utilizes "trimmed mean of mapped values" for normalization [29]. Similar to baySeq, EBSeq is based on empirical Bayesian analysis but employs median normalization [30]. NOISeq is only used with datasets that do not have replicates [24]. SAMseq employs a nonparametric statistical test that can handle outliers [31]. ShrinkSeq is used in studies with small sample sizes. Authors claim the algorithm performs better than edgeR and competes well with DESeq2 in terms of the false positives detected [32]. In brief, the most suitable tool for differential gene expression analysis is selected based on the dataset in question.

### 3.3. Downstream Analysis of the DEGs

The DEGs are ranked in order of their significance (*p*-value) and the $\log_2$ fold change; thresholds of these parameters are subjective to the study. Here, we present a few of the most used visualization methods and provide R-scripts to generate them (Figures 4 and 5, and Supplementary Document). There are three main ways to visualize different aspects of DEGs: MA plot, volcano plot, and heatmap presentation. A simple way to visualize the DEGs is the MA plot, in which the x-axis represents the number of the reads, and the y-axis represents the $\log_2$ fold change (MA stands for log ratio (M) and mean average (A)). This plot does not depict the statistical significance of the DEGs, and it is not frequently used in publications. Volcano plots depict the significance and expression levels of all of the genes

analyzed (Figure 4A). Another frequently used method to visualize RNA-seq data is the heatmap presentation (Figure 4B). The type of data presentation selected depends on the aspect of DEGs being highlighted.

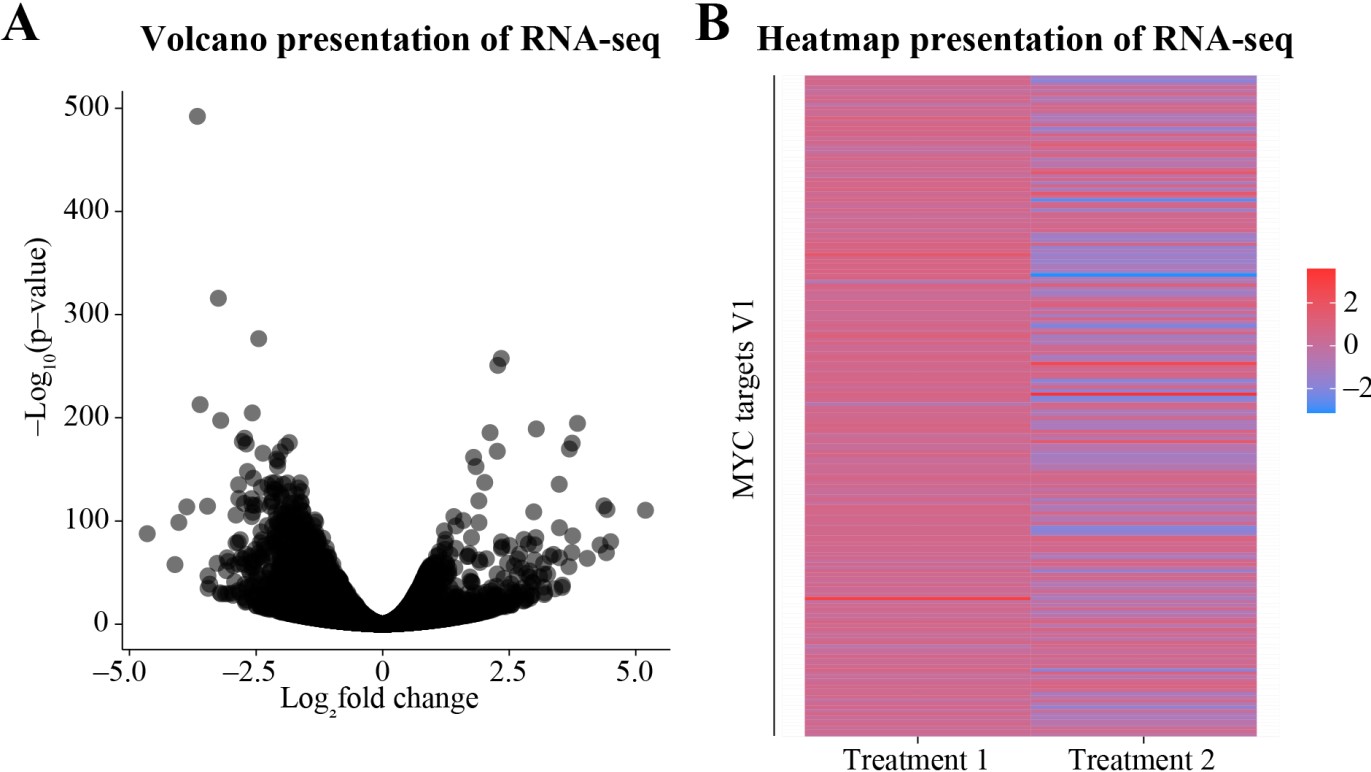

**Figure 4.** Presenting RNA-seq data using R. Refer to the supplementary document to find the relevant R scripts. Publicly available dataset (GSE116778) [33] was used to demonstrate how RNA-seq data can be visualized. (**A**) A volcano plot depicting the $\log_2$ fold change and the significance of the differentially expressed genes (DEGs) in treatment 1 (AT7519) [33]. Volcano plots are one of the most widely used modes of representation of DEGs as they depict both the fold change and the level of significance. (**B**) Heatmaps are a useful presentation method to compare multiple conditions. The heatmap here shows the $\log_2$ fold change in the MYC target genes from two treatment conditions (AT7519 and AT7519 + OSMI-2).

The DEGs can be further used for pathway enrichment and clustering to identify the biological processes affected. Online web servers such as Database for Annotation, Visualization and Integrated Discovery (DAVID) [34] and Enrichr [35] provide easy-to-use platforms for pathway and gene ontology (GO) enrichment. Cluego-plugin, which can be accessed through Cytoscape, groups the DEGs into clusters based on the GO term enriched for the categories [36]. Gene set enrichment analysis (GSEA) [37] has gained popularity in the past few years as it is a powerful analytical method to cluster and enrich the GO terms for the DEGs (Figure 5). GSEA is published as an open-source windows-based application and as an R-package. A more recent addition to the pathway enrichment tools is GeneWalk [38], a Python package that utilizes representation learning to identify regulator and moonlighting genes. The backend database and the statistical tests used to calculate the significance in the above-discussed programs differ. Therefore, multiple enrichment tools can be used for the same dataset for cross-validation and for a better understanding of the biological processes affected. This will enhance the certainty of the enriched GO terms; however, experimental validation is advisable despite the computational cross-validation.

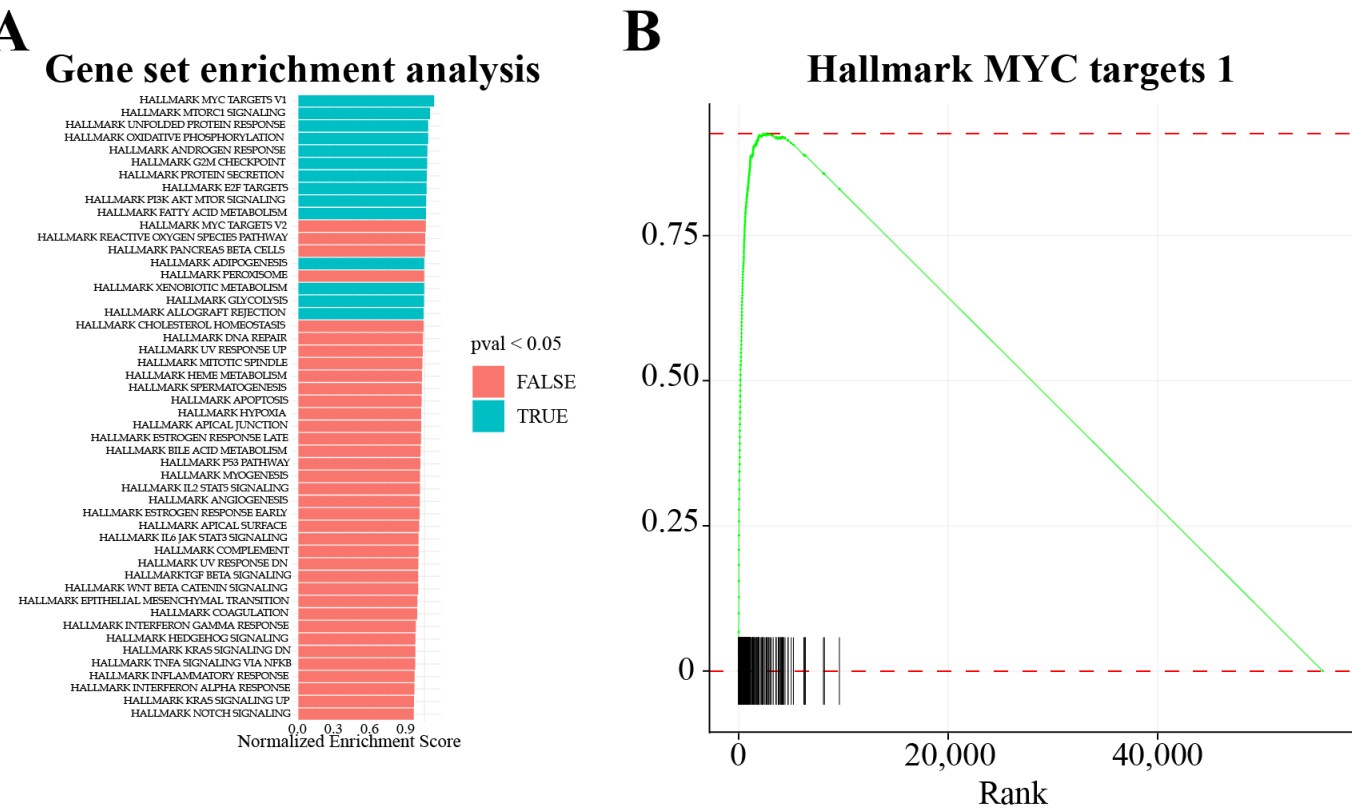

**Figure 5.** Gene set enrichment analysis (GSEA). Refer to the supplementary document to find the relevant R scripts. (**A**) All the DEGs from knockdown of O-GlcNAc transferase after 48 h in LNCaP cells were subjected to GSEA to identify the processes affected. The normalized enrichment score assigned by the GSEA algorithm is plotted and the color of the bar depicts the significance (blue: significant; red: non-significant) (data from GSE169090) [39]. (**B**) The enrichment score of the DEGs belonging to the significant pathway can be plotted using the GSEA enrichment plot. Here are plotted the MYC targets V1 Hallmark gene set for knockdown of O-GlcNAc transferase after 48 h in LNCaP cells. The X-axis shows the rank of the DEGs (calculated based on the $\log_2$ fold change and the $p$-value), and the Y-axis shows the enrichment scores.

### 3.4. Nascent RNA Sequencing Technologies

Capturing the actively transcribed mRNAs provides a direct measure of RNA polymerase II activity. In other words, nascent RNA sequencing technologies measure the active mRNA synthesis, not the overall mRNA abundance. In standard RNA-seq, the same absolute amount of RNA is sequenced for every sample. This means that the relative abundance of a given mRNA is evaluated regardless of if the gene is being actively transcribed.

Using biosynthetic metabolic labels, it is possible to measure the actively transcribing genes, and a number of tools have been developed for this purpose (Table 2). In 2005, Cleary et al. were among the pioneers of this approach, and they published a protocol to metabolically label RNA using 2,4-dithiouracil [40]. This protocol involves biotinylation of the labeled RNA, streptavidin-coated magnetic beads-based enrichment and analysis using microarrays. More recently, Core et al. (2008) presented a method based on massively parallel sequencing termed global run-on sequencing (GRO-seq). GRO-seq enables mapping of the position, amount and orientation of the transcriptionally engaged RNA polymerase II genome-wide [41]. Authors used 5-bromouridine 5′-triphosphate (Br-UTP), which is incorporated into the RNA. This RNA is further hydrolyzed and purified using beads coated with antibodies against 5-bromo-uridine. After nascent RNA cap removal and end repair, the eluted RNA undergoes reverse transcription to cDNA and is sequenced.

**Table 2.** Tools to measure active RNA synthesis. A myriad of techniques has been developed to analyze RNA synthesis in real time.

| Technique | Advantages/Comments | Limitations |
|---|---|---|
| Analysis of 2,4-dithiouracil labeled and enriched RNA using a microarray [40]. Description: the labeled RNA is biotinylated and enriched using streptavidin beads. The isolated RNA is analyzed using a microarray. | • Novel technology at the time; has since been replaced by sequencing-based tools. | • Dependent on the enrichment of 2,4-dithiouracil incorporated into RNA using affinity-based purification (biotin–streptavidin). • Microarray-based detection leads to high background signal. |
| GRO-seq (global run-on sequencing [41]). Description: labeled nucleotides (Br-UTP), are incorporated into the RNA. The RNA is then hydrolyzed and purified using antibody-coated beads. | • First metabolic labeling technique coupled to massively parallel sequencing. | • Isolation of nuclei may result in the loss of transcriptional regulators. • Affinity-based purification protocol is laborious and can result in the loss of signal of lowly expressed mRNAs. |
| PRO-seq (precision nuclear run-on sequencing [42]). Description: biotinylated NTPs are incorporated into the nascent mRNA, inhibiting transcription. 3′ end sequencing reveals the precise location of the stalled RNA polymerase. | • Allows precise detection of the active site of RNA polymerase engaged with its nascent RNA. | • Isolation of nuclei may result in the loss of transcriptional regulators. • Affinity-based purification protocol is laborious and may result in the loss of signal of lowly expressed mRNAs. |
| TT-seq [43] (Transient transcriptome sequencing). Description: label with 4-thiouridine, isolate RNA, fragment RNA, biotinylate and purify the labeled RNA, and sequence. | • Enables sequencing of the nascent RNA only. | • Affinity-based purification is laborious and may result in the loss of signal of lowly expressed mRNAs. • Labeling time is critical and has to be selected based on the scientific question. |
| SLAM-seq (Thiol (SH)-linked alkylation for the metabolic sequencing of RNA [44]). Description: labeling with 4-thiouridine, followed by alkylation which allows the nucleotide analog to be recognized as a cytosine. To measure the nascent mRNA synthesis the T > C conversion is measured. | • Affinity-based purification is not used. • The labeled mRNA transcripts are identified computationally. | • Labeling time is critical and has to be selected based on the scientific question. |

Precision nuclear run-on sequencing (PRO-seq) is an adaptation of GRO-seq, where the Br-UTP is replaced by biotinylated nucleotide triphosphates (NTPs) [42]. The incorporation of biotin-NTP inhibits transcription and 3′ end sequencing reveals the precise location of the active site of the RNA polymerase engaged with the nascent RNA. In 2016, Schwalb et al. used 4-thiouridine (4sU) as the starting point to develop transient transcriptome sequencing (TT-seq) [43]. In TT-seq, the nascent mRNA is labeled using the nucleotide analog 4sU for a very short period (5 min). As the transcripts are fragmented and enriched using streptavidin-coated magnetic beads before sequencing, only the actively transcribed genes are sequenced.

Over the years, metabolic labeling has gained popularity, which has led to the development of more sensitive methods that do not require enrichment of the labeled mRNA but rather rely on computational deconvolution. In 2017, Herzog et al. described thiol (SH)-linked alkylation for metabolic labeling of RNA (SLAM-seq) [44]. SLAM-seq is a chemistry-based RNA-seq technique that detects the 4sU incorporation at the single nucleotide resolution. In brief, 4sU is added to cells that incorporate the label to the actively transcribing mRNAs. The isolated RNA is alkylated by iodoacetamide, which renders the labeled site to be recognized as C during the library preparation. Finally, 3′mRNA sequenc-

ing technology is used. Development of the metabolic labeling methods has generated the need for specific computational tools to analyze the generated data.

### 3.5. Downstream Analysis of the Metabolically Labeled RNA

The tools used for standard RNA-seq can be used for nascent RNA-seq analysis; however, depending on the experimental setup, certain dedicated algorithms may be necessary. The quality check and data pre-processing are the same as presented for standard RNA-seq (Section 2) and are essential to perform.

For GRO- and PRO-seq, the alignment steps are the same as discussed above (Section 3.1) but the analysis of the aligned files is ideally performed using dedicated packages. groHMM is an R package that is used to analyze the aligned files from GRO- and PRO-seq [45]. In addition, HOMER [46] can be used to analyze GRO-seq data; however, groHMM outperforms HOMER in terms of the coverage of genic and intergenic regions, as well as in transcription unit accuracy for both short and long transcripts [45,47].

For SLAM-seq, the majority of the data analysis pipeline is developed to fit this particular approach. Identification of the nascent mRNA is based on the number of T > C conversions observed per transcript. To identify the transcripts with T > C conversions, a specific data analysis pipeline termed SLAM-dunk was developed [48]. The SLAM-dunk analysis outputs two major files: the tab-delimited count file, having the T > C conversion rates, and the filtered BAM file, which has only the labeled transcripts. These two files are used for downstream analysis. Using the T > C filtered BAM files and the conversion rates from the count file, the differentially labeled transcripts (DLTs) can be called. In practical terms, the DEG calling algorithms described above (DESeq2 and edgeR) can be used for calling the DLTs. The called DLTs can be visualized using similar tools as for the standard RNA-seq including volcano plots and heatmaps (Section 3.3).

Another tool to analyze SLAM-seq data is GRAND-SLAM (globally refined analysis of newly transcribed RNA and decay rates using SLAM-seq), a patented tool developed by the Erhard lab [21]. The tool requires the FastQ files to be aligned using STAR aligner (as it is a splice-aware aligner). The aligned BAM files must be converted into CIT format before inputting them into GRAND-SLAM. The main output table with all the information is saved in a tsv file. The two central parameters from the output are the read count and the total-to-new ratio. This tab-delimited file serves as the input to GrandR (an R package) to call the differentially labeled transcripts.

### 3.6. Analysis of Alternative Splicing

Splicing of mRNA is necessary to generate mRNA suitable for translation into proteins. Alternative splicing is a specific method of splicing in which, after translation, a cell generates alternative protein isoforms from the same gene. There are five major alternative splicing events: exon skipping (SE), retained intron (RI), mutually exclusive exons (MXE), alternative 5′ splice sites (A5SS), and alternative 3′ splice sites (A3SS) [49]. These alternative splicing events greatly increase the amount of potential isoforms for every given gene. The input files for the alternative splicing analysis are standard RNA-seq data files. In practical terms, through the analysis of DEGs and alternative splicing, the differential gene expression data can be integrated into the upstream processing. Alternative splicing analysis requires additional validation using orthogonal methods to confirm if the generated transcript is biologically relevant or if it is rapidly degraded.

MISO, rMATS, and SUPPA are the major tools used for calling the differentially spliced sites from the RNA-seq data. Mixture of isoforms (MISO), developed in 2010, was one of the first alternative splicing analysis tools [50]. MISO is a statistical model that estimates the expression of the alternatively spliced exons and isoforms and provides a confidence estimation. rMATS is a robust statistical method developed in 2014 [20], and, a year later, Alamancos et al. reported an alternative splicing calling algorithm termed SUPPA [51]. According to the authors, SUPPA is based on transcript abundance and is 1000 times faster than the other two algorithms discussed. Even though SUPPA is time-efficient, it requires

an additional step of calculating the transcript abundance, whereas rMATS can be run using the FastQ files directly. All of the algorithms mentioned above report the percent spliced in (PSI)- and confidence evaluation. The PSI index is the efficiency of splicing all the exons and retained introns genome-wide [52]. In other words, the PSI index of a gene is indicative of the intensity of its alternative splicing event and its inclusion or exclusion.

### 3.7. Single-Cell RNA-seq

Single-cell RNA sequencing (scRNA-seq) provides a higher resolution of cellular differences and a better understanding of the role of an individual cell in the context of the microenvironment (Figure 6A). The method is used to identify the different cell types in a sample, the cell cycle phases, and trajectory analysis (pseudo-time). scRNA-seq was initially developed by the Surani laboratory in 2009 [53]. The current scRNA-seq protocol involves encapsulating the single cells into unique barcode-containing droplets in a microfluidic device. Reverse transcription occurs in the droplets, and when the cDNA libraries are sequenced, all the cells are associated with a unique barcode.

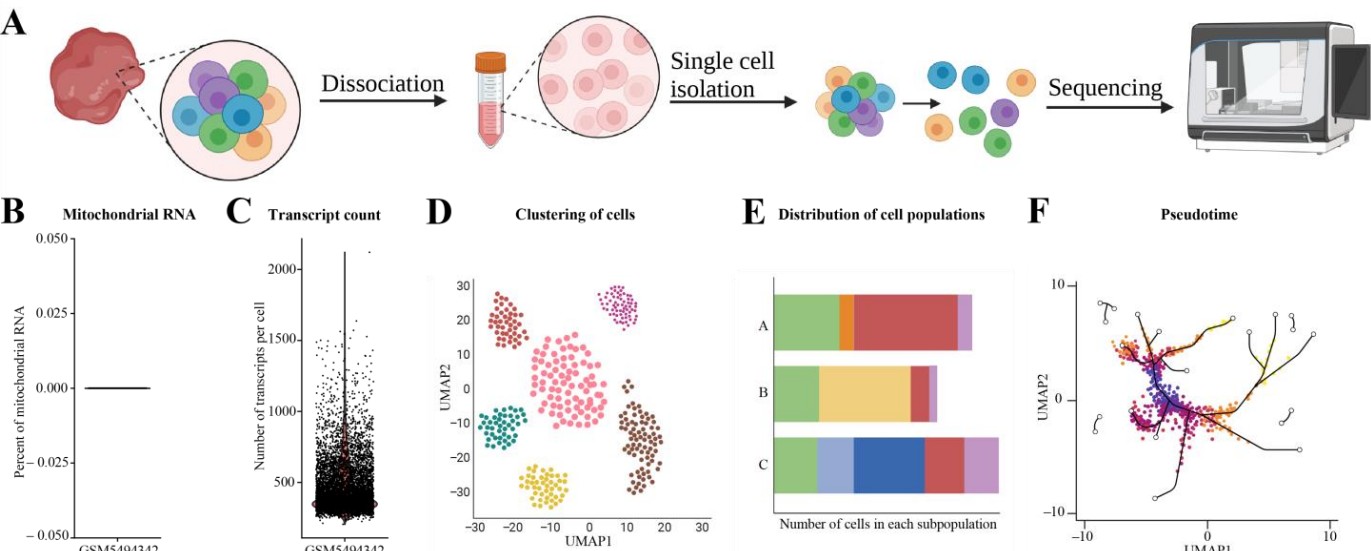

**Figure 6.** Single-cell RNA-seq (scRNA-seq) workflow, data quality evaluation, and visualization. (**A**) Workflow of scRNA-seq. (**B**) Evaluation of mitochondrial RNA contamination using publicly available dataset (GSM5494342). (**C**) Number of transcripts in each cell (we used publicly available dataset GSM5494342 to generate the plot). (**D**) Visualization of scRNA-seq data using UMAP. (**E**) Bar plot depicts the number of different cell types in the samples. (**F**) Example of how the pseudo-time visualization looks (example made using BioRender).

Once the reads are obtained, the first step is quality control. Low-quality reads and each read's adapter sequences are removed using the same tools used for bulk RNA-seq (discussed in Section 2.2). The quality of the data can be assessed based on the mitochondrial RNA content, which should be minimal (Figure 6B). High levels of mitochondrial RNA indicate that the cells have ruptured during the sample preparation. Another measure to assess the quality is the number of expressed genes/transcripts in each cell, which should be high and similar across the samples in the study (Figure 6C). Both the mitochondrial RNA content and expressed genes can be assessed by plotting the read count matrix using the Seurat package in R [54]. The next step is read alignment, which can be performed using the same tools as for the bulk RNA-seq. The widely used methods for read alignment are TopHat [16], STAR [17], HISAT2 [55], and Cufflinks [56]. After the alignment, reads mapping to the exonic region with high mapping quality are used to generate the gene expression matrix. As scRNA-seq data are highly noisy, it is necessary to normalize

the technical variabilities. The most frequently used approaches are RPKM and FPKM (discussed in Section 3.2).

The major applications of scRNA-seq data are to identify the cell subpopulation from the sample, differential gene expression analysis, and pseudo-time reconstruction. Seurat, an R package includes modules and functions to perform all these [54]. In addition, scRNA-seq data can be used to identify the subpopulations of the cells which express a certain gene signature. Commonly used means of visualizing the scRNA-seq data are violin-, UMAP-, and bar-plots, along with pseudo-time trajectory (Figure 6C–F). The benefit of scRNA-seq is the ability to discover sub-populations within the sample. However, scRNA-seq only detects a relatively small number of transcripts per cell, and to identify low-abundance transcripts, standard "bulk" RNA-seq is required.

## 4. Future Perspectives

The availability of sensitive and time-efficient algorithms for transcriptomics data analysis has enabled the scientific community to answer biological questions, which we could not even think of answering two decades ago. Within the past decade, the number of tools to analyze RNA-seq data has increased significantly and includes analysis of differential gene expression, alternative splicing, nascent RNA, and expression at the single-cell level. In addition, RNA-seq data can be used to discover fusion genes, and we refer the reader to the papers that evaluate these tools as these go beyond the major focus of this review [57–59].

Many of the tools to analyze RNA-seq data are not easy to use due to the technical requirements (high computing power, data storage, and computational skills), and the ability to use the command line-based tools. This calls for the development of automated data analysis pipelines with which minimal computational knowledge is needed to operate. These automated pipelines would ascertain that high-quality tools are available for the use by the entire scientific community, irrespective of the field of specialization. In this last section, we reflect on the next critical steps needed to further develop and utilize RNA-seq technologies.

### 4.1. Which Sequencing Technology Is Most Suitable for a Particular Experiment?

The scientific question dictates the correct transcriptional profiling approach. In the traditional RNA-seq, the same amount of RNA is sequenced, which will give a picture of the overall mRNA levels. This can be sufficient in certain cases, most typically when the researcher is comparing healthy versus unhealthy samples. Standard RNA-seq data can also be used to analyze alternative splicing, which is a powerful asset in particular experimental setups and will increase understanding of complex biological systems. However, in RNA-seq, utilization of the same absolute amount of RNA for sequencing is the gold standard approach but can lead to misinterpretation of the data, for example, when overall transcription is inhibited. In these experiments, the ability to measure active RNA synthesis is of critical importance. We summarize the key features of frequently used sample preparation methods in Tables 1 and 2 and compare their advantages and disadvantages. These tables are intended to be used as a reference to select the right approach for any particular study.

### 4.2. The Power of Metabolic Labeling

Analysis of the nascent RNA levels enables the identification of the immediate effects of experimentation on mRNA synthesis. In these experiments, the synthesized RNA is labeled and analyzed by sequencing using relatively complex experimental setups (GRO-seq and PRO-seq) or computationally decoded (SLAM-seq). Integration of the metabolic labeling techniques into new experimental setups will be powerful. For example, metabolic labeling of RNA using 4sU followed by immunoprecipitation of a splicing regulator or an RNA binding protein would enable tracking of the velocity of splicing in response to a

particular treatment. Essentially, a modified RIP-seq assay can be developed [60] and the data analyzed using SLAM-dunk.

### 4.3. Selecting the Right Data Analysis Pipeline

High throughput sequencing technologies are prone to varying levels of "noise", which can lead to misinterpretation of the data. This issue can be managed by focusing on the most significantly affected mRNAs or by validation using additional methods, such as another sequencing technology. Another, albeit less often utilized approach, is to apply multiple data analysis pipelines to call the differentially expressed/labeled mRNAs from the same dataset. We propose that the utilization of multiple data analysis pipelines, which rely on different mathematical models, is of high importance in scRNA-seq, particularly in scSLAM-seq [61,62]. The utilization of more than one data analysis pipeline can be used in such instances to increase confidence in the results generated.

**Supplementary Materials:** The following supporting information can be downloaded at: https://www.mdpi.com/article/10.3390/cimb45030120/s1.

**Author Contributions:** Both A.G. and H.M.I. have equally contributed to writing the final version of the article. All authors have read and agreed to the published version of the manuscript.

**Funding:** A.G. is supported in part by the Young Investigators' Grant (University of Helsinki), Orion Research Foundation sr, and K. Albin Johanssons stiftelse. H.M.I. is grateful for the funding from the Academy of Finland (Decision nr. 331324 and nr. 335902), the Jenny and Antti Wihuri Foundation, and Sigrid Juselius Foundation. Funding sources had no role in writing the report or in the decision to submit the article for publication.

**Institutional Review Board Statement:** Not applicable.

**Informed Consent Statement:** Not applicable.

**Data Availability Statement:** This is a review article, no new data was generated.

**Acknowledgments:** The authors wish to acknowledge CSC—IT Center for Science, Finland, for the computational resources. We are grateful to Rhiannon Newman for the comments on the manuscript.

**Conflicts of Interest:** The authors declare no conflict of interest.

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
