# Peer review of "Revealing the History and Mystery of RNA-Seq"

_cimb, doi:10.3390/cimb45030120_

Round 1

Reviewer 1 Report (Previous Reviewer 1)

The suggested corrections and additions where met.

Author Response

Thank you. We appreciate your comments, which significantly improved our manuscript.

Reviewer 2 Report (New Reviewer)

This manuscript from Aishwarya and colleagues presents a comprehensive review focusing on RNA-seq. Overall, this is a great review and should interest researchers who plans to start doing RNA-seq work. But the manuscript I received has some format issues. A lot fields are highlight in red and not sure why. One typo I noticed is in section 2.2, " In the sequencing reaction adapters allow the RNA-fragments", it should be DNA-fragments, correct?

Author Response

Thank you for a very positive review. Indeed, one of our goals was to write a review that makes it easier for new scientists to start doing RNA-seq work. All the text is now written with the same color (black). The corrections requested by the Reviewers are highlighted with the ‘track changes’ function for the ease of the Reviewers and the Editor.

Thank you for noticing the typo in section 2.2, we have now corrected this part and it reads: ‘In the sequencing reaction adapters allow the fragmented sample to bind to the lanes of the sequencer’.

Reviewer 3 Report (New Reviewer)

This review focuses on the application of current bioinformatic tools and analysis steps using RNA-seq data (including bulk, nascent, and single cell RNA-seq). In addition, high-level comparisons between these technologies are given. Given the broad spectrum, it is understandable that this review does not aim at being thorough, but as a resource for readers who are not familiar with RNA-seq to use a starting point. To this end, the review includes the most commonly used tools and data analyses, and even provides a very brief R script sample. Overall, this could serve as a useful resource for some readers in the community.

Here I include multiple concerns/suggestions in the order of where these first showed in the review:

1.     Figure 1 the left panel is repeated in Figure 2.

2.     Figure 1 the right panel only included alignment, although assembly is also discussed in the review.

3.     In addition to DE, transcription, and alternative splicing, there are some other common applications, e.g., gene fusion discovery, that could be included.

4.     As an overview for the paper, not just bulk RNA-seq, Figure 1 does not include nascent and scRNA-seq? Either update the figure or the text could make this more consistent.

5.     In 3.1, by “hypothetical transcriptome”, do you mean the contigs? Please clarify.

6.     Also, in the same section, in addition to space, BAM is also faster in run time.

7.     Between 3.1 (alignment) and 3.2a (normalized read/feature counts), you didn’t include tools for feature/read counts? Such as featureCounts. Or for the RPKM, FPKM etc normalized values, you didn’t discuss tools used for generating these data.

8.     In 3.5, you may want to cite a (review) paper for these alternative splicing patterns.

Author Response

Reviewer 4 Report (New Reviewer)

The manuscript entitled “Revealing the History and Mystery of RNA-Seq” by Gondane and Itkonen summarizes RNA-sequencing technology including its various specialized applications to analyze active transcription, splicing, and transcriptomes of heterogenous cell populations. Given the spectrum of applications and data output there is growing need to develop computational approaches that efficiently and accurately process these comprehensive data sets derived from the appropriate RNA starting material including the distinct methods used to prepare this nucleic acid population for sequencing. This review provides a logical and organized description of the current RNA-seq technologies used currently in molecular biology laboratories. Five figures and two tables are included in this review which effectively complement the information provided in the text. Overall, this is a well written and concise review paper containing pertinent information on RNA sequencing technologies and data analyses.

Minor concerns:

1. Although the review paper already has five figures the scRNA-seq section seems too brief and would benefit from a bit more detailed discussion along with a figure.

2. By adding an extra figure for the scRNA-seq the authors may consider combining Figures 4 and 5 into one figure with four panels (A through D) since these are related to each other.

Author Response

This manuscript is a resubmission of an earlier submission. The following is a list of the peer review reports and author responses from that submission.

Round 1

Reviewer 1 Report

The sentence p1, line 38-40; "With the advancements of the automated sequencing techniques, quantitating the gene expression level across diverse samples became possible." has to be specified

p2. first paragraph; the first human genome was not done by NGS but by Sanger.

The first figure which is not  mentioned in the text (figure is ok)?

p5, line 129BWA uses a refrence for alignment

Generelly a more elaborate description of what is seen in the figures would be appreciated.  

p8, line 257 Sentence should be reformulated "Splicing of mRNA is necessary to generate the productive mRNA suitable for translation,.."

Reviewer 2 Report

The manuscript entitled "Revealing the history and mystery of RNA-seq" describes the history and application of RNA-Seq. This manuscript describes this technology and process. The information in the manuscript is suitable for reading but isn't enough as an article review. The manuscript talks about titles published many years ago and here I couldn't find any novelty. 

An article review should have some ideas and suggestions for future studies and scientists which I couldn't understand here. 

Finally, I think the manuscript is not novel and attractive to the audience. 

Also, the text has a lot of Grammarly errors and major revision is needed. 
